# Comparison of Two *Schizophyllum commune* Strains in Production of Acetylcholinesterase Inhibitors and Antioxidants from Submerged Cultivation

**DOI:** 10.3390/jof7020115

**Published:** 2021-02-04

**Authors:** Jovana Mišković, Maja Karaman, Milena Rašeta, Nenad Krsmanović, Sanja Berežni, Dragica Jakovljević, Federica Piattoni, Alessandra Zambonelli, Maria Letizia Gargano, Giuseppe Venturella

**Affiliations:** 1Department of Biology and Ecology, Faculty of Sciences, University of Novi Sad, TrgDositejaObradovića 2, 21000 Novi Sad, Serbia; jovana.maric@dbe.uns.ac.rs (J.M.); nenad.krsmanovic@dbe.uns.ac.rs (N.K.); 2Department of Chemistry, Biochemistry and Environmental Protection, Faculty of Sciences, University of Novi Sad, Trg Dositeja Obradovića 3, 21000 Novi Sad, Serbia; milena.raseta@dh.uns.ac.rs (M.R.); sanja.beric@dh.uns.ac.rs (S.B.); 3Institute of Chemistry, Technology and Metallurgy, University of Belgrade, Njegoševa 12, 11000 Belgrade, Serbia; djakovlj@chem.bg.ac.rs; 4Laboratory of Genetics & Genomics of Marine Resources and Environment (GenoDream), Department Biological, Geological & Environmental Sciences (BiGeA), University of Bologna, Via S. Alberto 163, 48123 Ravenna, Italy; federica.piattoni@unibo.it; 5Dipartimento di Scienze e Tecnologie Agroalimentari, University of Bologna, Via Fanin 46, 40127 Bologna, Italy; alessandr.zambonelli@unibo.it; 6Department of Agricultural and Environmental Science, University of Bari “Aldo Moro”, Via Amendola 165/A, I-70126 Bari, Italy; marialetizia.gargano@uniba.it; 7Department of Agricultural, Food and Forest Sciences, University of Palermo, Via delle Scienze, Bldg. 4, 90128 Palermo, Italy; giuseppe.venturella@unipa.it

**Keywords:** *Schizophyllum commune*, submerged cultivation, acetylcholinesterase inhibition, antioxidant

## Abstract

In recent years, fungi have been recognized as producers of acetylcholinesterase (AChE) inhibitors, agents important for the prevention of Alzheimer’s disease (AD). This study aimed to examine the AChE inhibitory, the antioxidative and antibacterial activity of two different *Schizophyllum commune* strains that originated from Serbia (SRB) and Italy (IT). Submerged cultivation of grown mycelia (M) and fermentation broth (F) of ethanol (EtOH) and polysaccharide (PSH) extracts lasted for 7, 14, 21 and 28 days. For AChE activity Ellman method was performed, while for antioxidative activity, sevendifferent assays were conducted: DPPH, ABTS, FRAP, SOA, OH, NO together with total phenolic content. Antimicrobial screen, LC–MS/MS technique and FTIR measurements were performed. Different isolates exhibited different AChE activity, with PSH being the strongest (SRB, M, 28 days IC_90_ 79.73 ± 26.34 µg/mL), while in EtOH extracts, IT stood out (F, 14 days, IC_50_ 0.8 ± 0.6 µg/mL). PSH extracts (7 days) exhibit significant antioxidative activity (AO), opposite to EtOH extracts where 14 and 21days periods stood out. Only tw extracts showed antibacterial activity. Following LC–MS/MS analysis *p*-hydroxybenzoic and gallic acids were the most abundant phenolics. PSH extracts demonstrated remarkable results, making this study debut and introducing *S. commune* as a valuable resource of AChE inhibitors.

## 1. Introduction

Reactive oxygen species (ROS) represent the most potent free radicals since they can have a destructive effect on various cells and cause oxidative stress [1,2]. The imbalance between the production and quenching of free radicals may lead to a wide range of diseases, including neurodegenerative disorders such as Alzheimer’s (AD) [3]. Even though the exact cause of AD remains a question of debate, various studies suggest that oxidative stress plays a key role in an early response [2,3,4,5]. Furthermore, due to a lack of treatment to stop or reverse its progression, acetylcholinesterase inhibitors (AChI) have become an attractive research topic. AChI medications can help with memory symptoms and other cognitive changes by increasing the level of acetylcholine in the brain [6]. Hence, a wide range of organisms have been studied in order to find a biologically active compound that would be a successful AChI [7] and today, the only natural approved agents are alkaloids, such as galantamine and huperzine A [8]. Nevertheless, problems with bioavailability and numerous side effects pointed out that there is aneed for other AChI [7,8,9].

In recent years, mushrooms have been recognized as producers of non-alkaloid agents that inhibit acetylcholinesterase enzyme (AChE) [10,11] as well as the organisms which possess naturally occurring antioxidants [12]. Many structurally different polysaccharides with high antitumor activity and no side effects have been isolated from fungi [13], and it has been observed that they have the ability to improve memory and learning [14,15]. In addition, secondary metabolites, mostly phenolics, are considered to be responsible for antioxidant activity (AO) [10,16]. Submerged cultivation represents a controlled biotechnological process for providing valuable mycelial biomass and extracellular medium, which contain different organic compounds and may show different properties. Moreover, it can provide a faster route for antioxidant and exopolysaccharide production in basidiomycetes [17].

*Schizophyllum commune* (phylum Basidiomycota, order Agaricales, family *Schizophyllaceae*) is a widely distributed wood—decaying basidiomycete—that is present on all continents except Antarctica, but it is more common to be found in warmer regions with high humidity [18]. Thanks to the production of a neutral extracellular polysaccharide schizophyllan (Sonifilan, SPG), a β-1,3 glucan with β-1,6 branching and molecular weight of 450 kDa, that has shown significant antitumor, anticancer, immunomodulatory and anti-inflammatory effects [12,19,20], other benefits of this mushroom are being investigated [15]. Considering that environmental conditions in different geographical regions influence, to a great extent on the bioactive properties, the purpose of this study was to examine and compare AChE inhibitory and antioxidative activity, in vitro, of ethanol and polysaccharide extracts of two different strains of *S. commune* originated from Serbia (SRB) and Italy (IT). Both isolates were cultivated in submerged conditions in order to examine and compare the activity of mycelium biomass (M) and fermentation broth (F).

## 2. Materials and Methods

### 2.1. Biological Material

Two dikaryon strains of the wild-growing *S. commune* Fries 1815. (Ph. Basidiomycota, Cl. Agaricomycetes, O. Auriculariales, Fam. *Schizopyllaceae*) were collected near Bologna, Italy, in 2016 and in Zmajevac (Fruška Gora mountain) in Serbia in 2012. Mushroom identification was carried out by studying the fungal morphology macroscopically (color, shape, size and hyphae) and microscopically [21]. Mycelia wereisolated from the fruiting bodies of both isolates and cultivated at 26 °C, for 10 days, on malt agar (Torlak, Serbia). Mycelia of both isolates were deposited in fungal culture collection FUNGICULT, (https://www.pmf.uns.ac.rs/en/research/groups/profungi/), of the ProFungi laboratory at Department of Biology and Ecology-DBE, Faculty of Natural Sciences-PMF, University of Novi Sad-UNS. The isolates were referenced under the following numbers: 0043 and 0047, respectively, for *S. commune* SRB and IT.

### 2.2. Submerged Cultivation and Preparation of Extracts

Mycelia of SRB and IT strain from culture collection were cultivated on malt agar (Torlak, Serbia) at 26 °C for 12 days in a thermal incubator in the dark, each isolate in triplicate. A sterilized fermentation medium was prepared in Erlenmeyer flasks (500 mL narrow throat flasks), in which 5 plaques of pure culture mycelia, approx. 1 cm^2^ in size, were transferred into 100 mL of fermentation broth that contained 0.5 g peptone, 3.7 g glucose, 0.17 g maltose, 0.17 g fructose, 0.17 g xylose, 0.5 g yeast extract, 0.1 g K_2_HPO_4_, 0.05 g MgSO_4_x7H_2_O and 0.005 g vitamin B1. Incubation on a rotary shakerat 120 rpm, 26 °C and in the light (IKA KS 4000i control, Werke GmbH and Co.KG, Staufen, Germany) lasted for 7, 14, 21 and 28 days. Every 7 days, three replicates of SRB and IT isolates were filtered (diaphragm vacuum pump GM-0.5, HINOTEK Group Limited, China) and lyophilized at −80 °C under vacuum (Bio alpha, Martin Christ GmbH, Switzerland), F for 72h and M for 48h. Afterward, samples were grounded to a fine powder (IKA A11 basic, Germany) and kept in dark bottles at room temperature until further use. Based on the measured masses of mycelium and lyophilized filtrate, appropriate growth curves were made (Appendix A). The mass of each strain was derived from 1200 mL of fermentation medium (three replicates with 100 mL for each incubation time). Ethanol extracts (EtOH), M and F, were prepared by mixing 5 g of fungal material with 100 mL of 96% ethanol-EtOH (Zorka Pharma, Serbia) and stirring at 120 rpm at room temperature for 72 h. Afterward, extracts were evaporated (Buchi R-210, Switzerland) until the dry weight (d.w.) is reached and redissolved in 80% EtOH to achieve a concentration of 10 and 20% (*w/v*). Polysaccharide (PSH) extracts were prepared according to Ren et al. with some modifications [22]. The modifications refer to the centrifugation speed;more precisely, instead of 14,000 rpm, 12,000 rpm was performed.

### 2.3. Determination of AChE Activity

In vitrodetermination of AChE inhibitory activity of PSH and EtOH extracts was done according to Ellman et al. adapted for the use in 96-well microplates [23]. As an artificial substrate, acetylcholine iodide-AChI, originating from electric eel (Sigma-Aldrich, Chemie GmbH—Schnelldorf, Germany), was used for degradation of the AChE enzyme (Sigma-Aldrich, Merck Group, Germany). The reaction mixture contained 20 µL of the fungal extract, 150 µL of reagent A (Ellman’s reagent and AChI) and 50 µL of reagent B (AChE enzyme at 518 U/mL previously dissolved in phosphate buffer, pH 8.0), while donepezil at 1 mg/mL (Donecept, Zdravlje, Leskovac, Serbia) was used as a positive control. The absorbance was measured at λ = 412 nm on a 96-well plate reader (Multiskan Ascent, Thermo Electron Corporation, USA) in a total of 15 measurements with an interval of 1 min. The concentration range of analyzed extracts was from 10% to 0.01%. The percentage (%) of enzyme inhibition was calculated based on the following Equation (1):I_AChE_ (%) = (1 − A_sample_/A_control_) × 100%(1)
where A_sample_ and A_control_ stand for the absorbance of the tested and control samples, respectively. All measurements were performed in triplicate, and the results were presented as IC_50_ and IC_90_ values, with lower values corresponding to higher AChE activity of the sample. The reference inhibition time is a period of 10 min or 600 s, respectively.

### 2.4. Determination of Antioxidant Activity

For the in vitro evaluation of the AO of PSH and EtOH extracts, the scavenging effect on 2,2′-diphenyl-1-picrylhydrazyl (DPPH) and 2,2′-azinobis(3-ethylbenzthiazolin)6-sulfonic acid (ABTS) radicals, in addition to ferric reducing antioxidant power assay (FRAP assay) of extracts was determined. Additionally, the capability to neutralize superoxide anion (SOA, O_2_**^−^**), hydroxyl (OH^·^) and nitric oxide radical (NO^·^) was assessed for EtOH. Synthetic antioxidant propyl gallate (PG) was used as a positive control for RSC assays, and all assays were completed in triplicates. The results were presented as the mean values ± standard deviations (SD).2.4.1. DPPH Assay

The DPPH free radical scavenging assay was carried out according to the previously described procedure [24]. Briefly, the reaction mixture contained 60 µL of 90 µM DPPH reagent, 180 µL MeOH and 10 µL of the relevant *S. commune* extract (EtOH and PSH). The mixture was kept in the dark at room temperature for 30 min, and the absorbance was measured at λ = 515 nm (multiscan, Thermo Scientific, Waltham, MA, USA). The following Equation (2) was used for the calculation of radical scavenging capacity (RSC):RSC _(DPPH)_ (%) = (1 − A_sample_/A_control_) × 100%(2)
where A_sample_ and A_control_ stand for the absorbance of the tested extracts and control. After regression analysis, results were presented as IC_50_ values (µg/mL) (the concentration of the test substance at which 50% of the radicals is neutralized).

#### 2.4.1. ABTS Assay

The ABTS test, based on spectrophotometric monitoring of the transformation of the blue-green colored cation radical ABTS^+^ into its neutral, colorless form, was carried out according to Arnao et al. [25]. ABTS^+^ was produced directly by reacting 7 mM ABTS solution with 2.45 mM K_2_S_2_O_8_, and after incubation (12–16 h in the dark at the room temperature) on the day of the assay, ABTS solution was diluted with EtOH to achieve absorbance between 0.800and 0.900 at λ =734 nm. 10µL of fungal extract (EtOH and PSH) was added to 290 µL of ABTS solution and mixed. The sample absorbance was read at λ = 734 nm after a 5-min incubation at room temperature, while Trolox was used to calculate the standard curve. The ABTS radical scavenging activity was expressed as mg of Trolox equivalents per g dry weight (mg TE/g d.w.).

#### 2.4.2. FRAP Assay

FRAP assay was done by the method of Benzie and Strain [26]. The fresh FRAP reagent consists of 10 mmol/LTPTZ solution in 40 mmol/L HCl, 0.02 mmol/L FeCl_3_x6H_2_O and acetate buffer (pH 3.6) in ratio 10:1:1. For the assay, 10 µL of each extract was mixed with 225 µL of FRAP reagent and 22.5 µL of dH_2_O. The absorbance was measured after 6 min at 593, while ascorbic acid (AA) was used to calculate the standard curve. Reducing power of analyzed fungal extracts (EtOH and PSH) was expressed mg AA equivalents per g dry weight (mg AAE/g d.w.).

#### 2.4.3. Superoxide Anion Radical Scavenging (SOA) Assay

The antiradical O_2_^−^ the activity of tested EtOH extracts was evaluated by a previously described method [27], based on measuring extract the ability to neutralize these radicals formed during aerobic reduction of NBT by NADH in the presence of phenazine methyl sulfate. The standard antioxidant PG was used as a positive control, and the neutralization potential of RSC_O__2-_was calculated according to the formula in Section 2.4.1. The results were expressed as the mean of the three IC_50_ values obtained ± SD (μg/mL).

#### 2.4.4. OH Assay

RSC activity of OH^·^ was determined by the modified method of Halliwell and Gutteridge [28]. 100 µL H_2_O_2_, 100 µL FeSO_4_ and 100 µL 2-deoxyribose-D-ribose were mixed with different concentration of EtOH fungal extract (10 µL) and 2.7 mL of phosphate buffer (pH 7.4). Samples were incubated for 60 min at 37 °C, and 200 µL of EDTA (ethylenediaminetetraacetic acid) and 2 mL of TBA reagent (5.2 mL perchloric acid, 1.5 g thiobarbituric acid and 60 g trichloroacetic acid diluted in 400 mL dH_2_O) were added in a mixture. The absorbance of the reaction mixture (pink complex) was noted at λ = 532 nm, and results were expressed as IC_25_ values ± SD (µg/mL).

#### 2.4.5. NO Assay

The concentration of nitrite ions was determined according to Green et al. [29]. The reaction mixture contained 15 µL of extract, 250 µL 10 mmol/L sodium nitroprusside and 250 µL phosphate buffer (pH 7.4). After incubation (90 min at room temperature and constant light), 500 µL of Griess reagent (a mixture of 0.2% solution of N-(1-naphthyl)-ethylenediamine dihydrochloride and 2% solution of sulfanilamide in 4% phosphoric acid) was added. The absorbance was measured at λ = 546 nm, and RSC was expressed as IC_25_ ± SD (µg/mL).

### 2.5. Antibacterial Activity

In vitroantibacterial susceptibility, assays were performed using 2 Gram-positive (*Staphylococcusaureus* and *Bacillus cereus*) and Gram-negative (*Escherichia coli* and *Pseudomonasaeruginosa*) bacteria strains from American Type Culture Collection (ATCC). Antimicrobial screen included agar-well diffusion and disc-diffusion assays in the evaluation of the antibacterial activity of EtOH fungal extracts of stock concentration, 10% (*w/v*). Evaluation of activity was performed according to the CLSI procedure [30] and the previously described method [16]. Evaluation of minimal inhibitory concentration (MIC) and minimal bactericidal concentration (MBC) was performed by the two-fold dilution susceptibility method according to the CLSI procedure [31] and Karaman et al. [32]. All tests were carried out in triplicate.

### 2.6. Determination of Total Phenolic Content

Determination of the total phenolic content (TP) was performed by the method of Singleton et al. [33]. Extracts were examined in the concentration of 0.625 to 50 mg/mL, while gallic acid (standard compound) was prepared in ten concentrations ranging from 0 to 100 μg/mL 25 μL of each extract or standard solution, except in blank probe where only the solvent was used (80% EtOH), was added to 125 μL of 0.1 mol/L Folin–Ciocâlteu reagent (FC) and mixed with 100 μL of sodium carbonate (7.5%) after 10 min. Absorbance at λ = 760 nm was read after 2 h incubation time. The phenolics concentration was determined by comparison with the standard calibration curve of gallic acid, and results were presented as a mean value of triplicated ± SD. The total phenol value was expressed as mg of gallic acid equivalents (GAE) per g of dry weight.

### 2.7. Fourier Transform Infrared Spectroscopy Analysis (FTIR)

FTIR spectra of the EtOH and PSH spectra were recorded on a Thermo-Nicolet Model 6700 spectrophotometer (Thermo Scientific, USA) equipped with Smart Orbit (Diamond) ATR accessory and OMNIC 7.3 software. Spectra were recorded in the 400–4000 cm^−1^ wave number range.

### 2.8. Hydrolysis

For analysis of the monosaccharide composition of PSH extracts, paper chromatography was applied. Each lyophilized PSH extract (2 mg) was hydrolyzed with 2 M TFA (2 mL) in a sealed tube at 100 °C overnight. Samples were concentrated by evaporation under reduced pressure (below 50 °C) to dryness. Residual TFA was removed by two evaporation cycles with 0.5 mL of isopropanol, and the final residue was dissolved in 0.02 mL ofdistilled water and analyzed by PC on Whatman No.1 chromatography paper (descending method) in the solvent system ethyl-acetate:pyridine:water (10:4:3 *v/v*/*v*). Components were visualized with alkaline silver nitrate [34]. D-Galactose, D-glucose, D-mannose, D-xylose, and D-ribose were used as standards.

### 2.9. LC–MS/MS Analysis

The quantification of the selected phenolic compounds in PSH and EtOH extracts of *S. commune* isolates was carried out using the slightly modified LC–MS/MS method by Orčić et al. [35]. To obtain the high selectivity and sensitivity, the selected reactions monitoring (SRM) acquisition mode was used since only ions specific to the targeted analytes were monitored. Extracts and standards were analyzed using Agilent Technologies 1200 Series high-performance liquid chromatography coupled with Agilent Technologies 6410A Triple Quad tandem mass spectrometer with electrospray ion source and controlled by Agilent Technologies MassHunter Workstation software Data Acquisition (ver. B.03.01). Compounds were separated on Zorbax Eclipse XDB-C18 (50 mm × 4.6 mm, 1.8 m) rapid resolution column held at 50 °C. Mobile phase was delivered at flow rate of 0.5 mL/min (instead of 1 mL/min) in gradient mode (0 min 30% B, 12 min 70% B, 18 min 100% B, 24 min 100% B, re-equilibration time 6 min, instead of 0 min 30% B, 6 min 70% B, 9 min 100% B, 12 min 100% B, re-equilibration time 3 min). Eluted compounds were detected by MS, using the ion source parameters as follows: nebulization gas (N_2_) pressure 40 psi, drying gas (N_2_) flow 9 L/min and temperature 350 °C, capillary voltage 4 kV, negative polarity. Data were acquired in dynamic MRM mode, using the optimized compound-specific parameters (retention time, precursor ion, product ion, fragmentor voltage, collision voltage). For all the compounds, peak areas were determined using Agilent MassHunter Workstation software Qualitative Analysis (ver. B.03.01). Calibration curves were plotted, and concentrations of samples calculated using the OriginLabs Origin Pro (ver. 9.0) software.

### 2.10. Statistical Analysis

All assays were performed in triplicate. The results were expressed as mean ± SD. IC_50_ and IC_25_ values were determined by the linear regression analysis of RSC (Origin Pro 2016 software, version 9.3). The data that have a normal distribution were subjected to two-way analysis of variance (ANOVA)and multivariate analysis of variance (MANOVA). Tukey’s test was used to determine significant differences (*p* < 0.05) between the extracts. Nonparametric Friedman tests and posthoc LSD tests were used for data that do not have a normal distribution. The correlation between antioxidant capacity and total phenols content was established using the Pearson’s product-moment correlation (normally distributed data) and Spearman correlation (data that do not have a normal distribution). The statistical analysis was performed using the IBM SPSS Statistics software version 22.0 for Windows.

## 3. Results

### 3.1. Inhibition of AChEEnzyme

The strongest potency was observed for PSH extracts of *S. commune,* where the highest concentration range (1%) overtakes IC_90_ (Figure 1A). Results of AChE inhibition activity (AChEI) of EtOH extracts were presented as IC_50_ value, owing to the lower inhibition than PSH, which are presented as IC_90_ values (Figure 1B). The IC_50_value of positive control (donepezil) was 87.92%.

Different strains of *S. commune* (IT, SRB) showed different AChEI activity. The PSH extracts of the SRB strain showed higher activity compared to the IT strain, while in EtOH extracts, the opposite was obtained (Figure 1). Among PSH extracts, the highest activity exhibited M of SRB strain after 28 days of incubation (IC_90_ 79.73 ± 26.34 µg/mL), while among EtOH, the strongest extract was F of IT strain after 14 days of incubation (IC_50_ 0.8 ± 0.6 µg/mL).

By comparing the inhibitory activity of M and F, higher AChEI activity was observed in the F within EtOH extract, while in PSH extracts M was stronger. Among EtOH extracts, differences in activity were related to the incubation period, and the strongest activity was shown after 14 days of incubation, apart from F extract originating from SRB, which showed the strongest activity after 28 days of incubation. PSH extracts did not show a statistically significant difference (ANOVA, *p* < 0.05) in inhibition of AChE when it comes to the incubation period, as opposed to EtOH extracts.

### 3.2. Determination of Antioxidant Activity

#### 3.2.1. DPPH Assay

All tested samples exhibited antiradical activity, with PSH being more active (Table 1). Among PSH extracts, the highest ability to capture DPPH radicals was shown by M after 14 days (IC_50_ 14.45 ± 6.83 for IT) and 7 days of cultivation (IC_50_ 18.92 ± 6.12 for SRB), while within F extracts those incubated 7 days distinguished (IC_50_ 16.03 ± 4.30 µg/mL for IT isolate and IC_50_15.76 ± 0.63 µg/mL for SRB isolate, respectively).

For EtOH extracts, the AO of F increased linearly from the 7th to the 21st day when it reached its highest activity (IC_50_ 55.43 ± 1.89 µg/mL for IT isolate and IC_50_ 55.96 ± 1.31 µg/mL for SRB isolate, respectively), and then on the 28th day, it began to decline for both isolates. Among M extracts, the analyzed antiradical activity had variations, but nevertheless, on day 14, they proved to havethe best ability to neutralize free radicals (IC_50_ 49.34 ± 0.65 µg/mL for IT and IC_50_ 74.65 ± 1.74 µg/mL for SRB, respectively).

#### 3.2.2. ABTS Assay

All analyzed samples (PSH and EtOH) showed the ability to “capture” ABTS radicals, with the PSH extracts being more active (Table 1). When we compare the two strains, F extracts were stronger in IT isolate and M extracts in SRB isolate. Overall, M extracts exhibited stronger activity compared to the F in both isolates.

#### 3.2.3. FRAP Assay

The ferric reducing ability of PSH and EtOH extract for both isolates was evaluated and compared (Table 1). By comparing all extracts, IT strain was more effective, with the exception of one SRB F extract (EtOH, 14 days), which exhibited the greatest activity (107.86 ± 12.81 mg AAE/g d.w). In all samples, F extracts showed a higher FRAP value than M. Moreover, in both strains, PSH F extracts demonstrated lower activity than EtOH F, as opposed to M extracts, where PSH extracts were stronger. According to statistical analysis (ANOVA, *p* < 0.05) incubation period did not increase the reduction potential of the extracts.

#### 3.2.4. SOA Assay

All tested EtOH extracts showed the ability to neutralize O_2_**^−^**. SRB strain was more potent compared to IT, while F extracts of both strains showed much higher antiradical activity (Table 2).

#### 3.2.5. OH Assay

EtOH extracts of *S. commune* (IT and SRB) showed the best OH^•^ neutralizing activity after 28 days of incubation (IC_25_ 10.80 ± 0.54 µg/mL for F IT, IC_25_ 89.69 ± 0.38 µg/mL for F SRB and IC_25_ 85.57 ± 0.12 µg/mL for M SRB), apart from M extract of IT strain which displayed the highest activity after 7 days (IC_25_ 20.20 ± 1.36 µg/mL) (Table 2). Summarized, extracts from IT strain were more effective.

#### 3.2.6. NO Assay

Among analyzed EtOH extracts, both isolates demonstrated moderate radical scavenging capacity, while some extracts were inactive or the activity of analyzed samples waslower than IC_25_ (Table 2).

### 3.3. Determination of Antibacterial Activity

Studies on the MIC and MBC for EtOH extracts demonstrated that only two extracts among all from SRB strain exhibited antibacterial activity, while IT strain was not active. The F extract of SRB strain after 21 days of cultivation demonstrated the highest antimicrobial activity (MIC and MBC < 0.31%) against *E. coli*, *B. cereus* and *S. aureus*, while the lowest was exhibited against *P. aeruginosa* (MIC and MBC 5%). F extract of SRB strain after 14 days of incubation demonstrated antibacterial activity (MIC 0.62% and MBC 0.31%) only against *E. coli.*

### 3.4. Total Phenol (TP) Content and Correlation Analysis

TP content is presented in Table 2. The amount of TP in both strains was high and rather similar; nonetheless, SRB extracts showed a higher value (84.60 ± 1.64 mg GAE/g d.w. for F and 82.62 ± 0.99 mg GAE/g d.w. for M, respectively). In addition, F extracts of both strains demonstrated higher TP content compared to M. Correlation analysis was performed to find out how AO of extracts is related to their TP content. A negative correlation between antioxidant properties and TP content was noticed in the ABTS assay of SRB extracts (r^2^ –0.79), while a close correlation (*p* < 0.05) in OH assay in both isolates (r^2^ 0.81 for IT and r^2^ 0.71 for SRB) was obtained. In addition, a positive correlation was detected between FRAP assay and TP content (r^2^ 0.69 for IT).

### 3.5. Chemical Characterization of Extracts

#### 3.5.1. FTIR Analysis

The spectra of all PSH samples investigated (Figure 2 (A–D) showed peaks characteristic for the presence of predominantly polysaccharides molecules, small quantities of protein and some aromatics. In all samples, the intensive sharp absorption at 1078–1080 cm^−1^ was characteristic of the presence of β-glucan due to O-substituted glucose residues [36]. The strong absorption at about 1050 cm^−1^ in each sample indicated that these polysaccharides had pyranose rings [37]. An intensive absorption band with a maximum at 1640 cm^−1^ corresponded to the characteristic frequency of protein as well as to the bending vibration O-H of associated water. This strong vibration was overlapped by specific absorption of aromatics (C=C and C=O stretch vibrations) and indicated the presence of phenolic compounds [38]. Frequencies at region 1410–1310 cm^−1^ corresponded to O-H groups of phenolics, too. In the anomeric region (950–700 cm^−1^), the characteristic weak absorption at 890 cm^−1^ in all samples corresponded to the presence of β-glycosidic linkages (C1-H deformation mode) [39]. Furthermore, the additional weak frequency at 935 cm^−1^ in B and C samples was also related to β-glycosidic bonds [40].

The FTIR spectrum of the EtOH extract (Appendix A) showed absorption bands characteristic for the carbohydrate polymers as well as proteins and polyphenolic compounds [41]. The bands in the region 1310–1410 cm^–1^ correlate with the valence vibrations of the OH groups of phenolic compounds. Among F samples of SRB extracts, the intensity of individual absorption bands was different. Namely, the intensity of bands in the region 1000–1200 cm^−1^ related to valence vibrations of CO and CC polysaccharide structures is reduced compared to the intensities of other absorption peaks. This indicates a decrease in the proportion of carbohydrate polymers relative to other compounds present in the mixture (e.g., phenols). The absence of the characteristic ester band in the region 1730–1740 cm^−1^ in all analyzed FTIR spectra indicates that the examined extracts do not contain uronic acids as part of their structural characteristics.

#### 3.5.2. Monosaccharide Composition

The presence of a large amount of D-glucose with smaller amounts of D-galactose and traces of D-mannose was confirmed in investigated hydrolysates by PC.

#### 3.5.3. LC–MS/MS Analysis

In all examined PSH and EtOH, among 45 investigated phenolics, 4 compounds were quantified using LC–MS/MS procedure (Table 3). Other phenolic compounds were above the limit of detection but lower than the limit of quantification (Appendix A). Three hydroxybenzoic acids and one cyclohexane carboxylic acid were detected. *p*-Hydroxybenzoic acid was detected in all investigated samples, while richer content was found in IT isolates (64.92 µg/g in total). The most abundant compound was gallic acid found in SRB F extract (75.77 µg/g). In comparison, EtOH extracts contained most of the analyzed phenolic acids, whereas F extracts showed higher phenolic content than M (167.55 µg/g in total).

## 4. Discussion

Inhibition of AChE enzyme was determined and compared for PSH and EtOH extracts of both investigated strains of *S. commune* (Figure 1). All PSH extracts, both M and F, exhibited significantly strong activity, hence expressed as IC_90_, which is comparable to the activity of a commercially approved AD drug donepezil (87.92%). Though some subtle distinction exists, the FTIR spectrum of PSH extracts (Figure 2) indicates that there are no significant differences in the composition of the F and M, owing to the mixture of polysaccharide fraction, probably schizophyllan structure [42], which is further expressed by their very similar AChE inhibitory activity.

Among EtOH extracts, stronger AChEI was observed for IT F extracts after 14 days of cultivation, while among PSH extracts, SRB M expressedgreater inhibition potential after 28 days of incubation. Furthermore, the incubation period did not show a statistically significant difference in PSH extracts (ANOVA, *p* < 0.05), while among EtOH incubation period had a greater impact (*p* = 0.04)on the AChEI activity, with 14 and 28 days being the highest (Figure 1B).This can be explained by the highest production of secondary metabolites, including phenolics in submerged cultivation in the stationary phase of cultivation when nutritive components are limited LC-MS/MS analysis (Table 3) revealed that within strongest EtOH extracts beside *p*-hydroxybenzoic acid, a high amount of quinic acid were found. Though in a previous study, quinic acid did not inhibit the AChE enzyme [43], it was shown that *p*-hydroxybenzoic acid exhibited lower inhibitory activity compared to 2-hydroxybenzoic acid [44]. The concentration of esculetin, a derivative of coumarin, in all examined submerged extracts of *S. commune* was higher than the limit of detection, while in *Coprinus comatus,* its presence in both mycelia and filtrate was recorded in a higher amount using HPLC-MS/MS procedure [45]. Ali et al. [46] demonstrated strong AChEI activity of this compound isolated from plants, and its AO was reported [44]. On the other hand, polysaccharides originating from mushrooms could be significant in AD therapy since in vivo experiments indicate that they might improve learning and memory [47]. Furthermore, the FTIR spectrum of PSH and EtOH extracts (Figure 2 and Appendix A) showed the presence of predominantly polysaccharide molecules next to proteins and phenolic compounds. Shizophyllan, β-1,3 beta-glucan with β-1,6 branching, a potent polysaccharide found in *S. commune*, may play an important role since PSH extracts exhibited high AChE inhibitory activity, opposite to PSH samples for *Trametes versicolor*, extracted after the same procedure, where activity was not detected [10]. In addition, EtOH samples demonstrated lower AChE inhibitory activity (44.35%) [10], compared to our results (Figure 1), while Abdullah et al. [48] recorded weak AChEI activity of hot water extracts of *S. commune* (IC_50_ 0.320 ± 0.070 mg/mL).

AO of PSH and EtOH extracts was evaluated in vitro using standard antioxidant assays (Table 1 and Table 2). Based on Table 1, there are statistically significant differences (ANOVA and Friedman, *p* < 0.05) among PSH and EtOH samples from SRB in DPPH and FRAP assays. EtOH samples from OH assay stands out since in NO and SOA tests, except F extracts, much lower antiradical activity was recorded (Table 2). The highest activity of F extracts (IT, 28 days) showed significant value (IC_25_ 10.80 ± 0.54 µg/mL) while among SRB samples, the activity of F and M (IC_25_ 89.69 ± 0.38 µg/mL andIC_25_ 85.57 ± 0.12 µg/mL, respectively) for the same incubation period was almost eight times lower.

Results of OH assays for EtOH extracts are comparable with TP content since a high positive correlation was obtained (r^2^ 0.81 for IT and r^2^ 0.71 for SRB). Moreover, these results suggest the potential use of this mushroom for the prevention of oxidative stress since OH is physiologically active and toxic [28]. DPPH assay did not show a statistically significant difference in correlation with TP content, although previous data from submerged methanol samples of *Coprinus* species demonstrated a positive correlation, which may be due to the application of different solvents [49].EtOH and MeOH extracts of *S. commune* showed three times lower anti-DPPH activity (IC_50_ 153.1 ± 0.02µg/mL) [48], while EtOH extract of the same species from India demonstrated two times higher RSC IC_50_ 18.56 µg/mL [50], compared to the strongest reported herein for EtOH extracts (IC_50_ 49.34 ± 0.65 µg/mL). EtOH extracts from submerged fermented mycelium (10 days) demonstrated stronger scavenging capacity as reported by Tripathi and Tiwary with IC_50_ 31.40 ± 0.05 µg/mL [51]. In other studies, lower scavenging capacity and reducing the effect of fruiting body extracts were recorded as well, while TP content varied from significantly reduced value (1.72 ± 0.05 mg GAE/g d.w. and 2.99 ± 0.04 mg GAE/g d.w.) to twice the lower value (41.07 ± 0.60 mg GAE/g d.w.) [52,53,54]. Nevertheless, submerged extracts of *S. commune* from Thailand showed higher DPPH activity (inhibition of DPPH ranged from 34.29 to 81.00%) with fermentation broth being twice as active [55], which is in accordance with our study and refers to the production of secondary metabolites during submerged cultivation. Given what has been said, a higher amount of TPcontent in F extracts (Table 2) quantified by LC–MS/MS (Table 3) indicates that fungal cells represent small bio-factories since various classes of metabolites can be produced during the same procedure of submerged cultivation. In comparison, PSH and EtOH extracts of *T. versicolor* exhibited RSC activity 50 times lower [10], which impliesthat besides phenolic compounds, primary metabolites, including polysaccharides, may play a role in AO.

LC–MS/MS procedure of extracts from the fruiting body of this mushroom species detected only protocatechuic and hydroxybenzoic acids in higher amounts [56], while lower amounts of various phenolic compounds were detected (Appendix A). This quantitative and qualitative variability is expected considering the different origins of fungal growth (mycelia, extracellular media) during submerged cultivation can produce different classes and concentrations of metabolites. According to growth curves, SRB strain (Appendix A) has rapid growth and reached the critical point of entry in secondary metabolism much earlier (14 days) than IT strain, when a very high concentration of gallic acid was observed. This can explain high DPPH and FRAP activities for SRB strain as well [57]. IT strain expressed critical points between 21 and 28 days (Appendix A) when different phenolic compounds were detected. All confirmed phenolic compounds (Table 3) are proven antioxidants [58,59], whereas LC–MS/MS analysis demonstrated higher phenolic content in EtOH extracts. Hence, the observed abilities of EtOH extracts may be explained by the presence of secondary metabolites, while primary metabolites may be responsible for high RSC activity among PSH extracts.

Antibacterial activity was very low as only two extracts showed activity. In a previous study, methanolic extract of culture filtrate incubated in submerged conditions for 10 days was active against two Gram-positive bacteria (*S. aureus* and *B. subtilis*) [51], which is in accordance with results from this study, since only F extracts of SRB strain showed moderate antibacterial activity.LC–MS/MS analysis and literature data both imply that gallic acid, protocatechuic acid and esculetin may be responsible for antibacterial activity since these compounds can inhibit the growth of *E. coli* [59,60,61,62]. Nonetheless, extracellular pigment melanin from *S. commune* was active against *E. coli* as well [63], suggesting a possible synergy effect.

Findings from this study suggest that different compounds, both primary and secondary, could be responsible for inhibition of AChE enzyme as well as for detected AO and antibacterial activities that strongly point out to a likelihood of synergistic activity of compounds in the extracts [16]. Based on observed AChEI activity, different geographical locations (strain origin—IT, SRB) from which the fungal strains were collected may affect the activity and the production of various metabolites probably due to different environmental factors that contribute to the biochemical patterns of the specific genotype (strain).

## 5. Conclusions

To the best of our knowledge, this is the first report describing the AChE inhibitory activity of PSH extracts of submerged extracts of *S. commune*. Most of the analyzed submerged samples exhibited AO, which is in direct relation to the characteristics of strain’s growth as it can point to the moment of entry in secondary metabolism, while in AChEIactivity, PSH extracts demonstrated remarkable results and represent first data for this species. Taken all together, PSH extracts obtained significant AO within samples incubated for 7 days, opposite to EtOH extracts where a 14- and 21-day cultivation period stood out, which was in accordance with LC–MS/MS phenolic analysis. Differences in activities between two isolates and different types of extracts, together with variations of phenolics content, indicate the important role of both genetic and environmental factors in metabolite production, as well as possible synergistic effect of primary and secondary metabolites. This study introduced *S. commune* as a potential natural antioxidant producer and indicated that the submerged fermentation might serve as an effective alternative method for producing bioactive agents, e.g., pharmaceuticals and nutraceuticals. In summary, this ubiquitous mushroom species should be considered as a valuable alternative source for future palliative therapy of AD.

## Figures and Tables

**Figure 1 jof-07-00115-f001:**
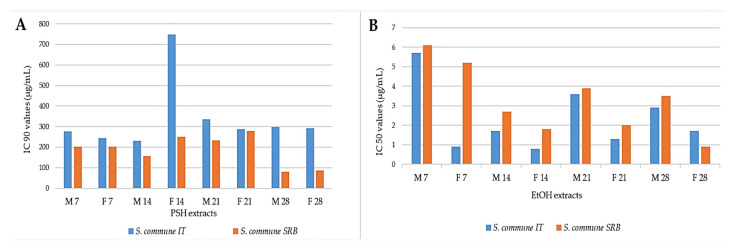
Comparative pre-presentation of acetylcholinesterase (AChE) inhibitory activity of two different isolates (Italy (IIT) and Serbia (SRB)) of *S. commune*:(**A**) inhibitory activity of submerged polysaccharide (PSH) extracts presented as IC_90_ values (µg/mL); (**B**) inhibitory activity of EtOH extracts presented as IC_50_ values (µg/mL).

**Figure 2 jof-07-00115-f002:**
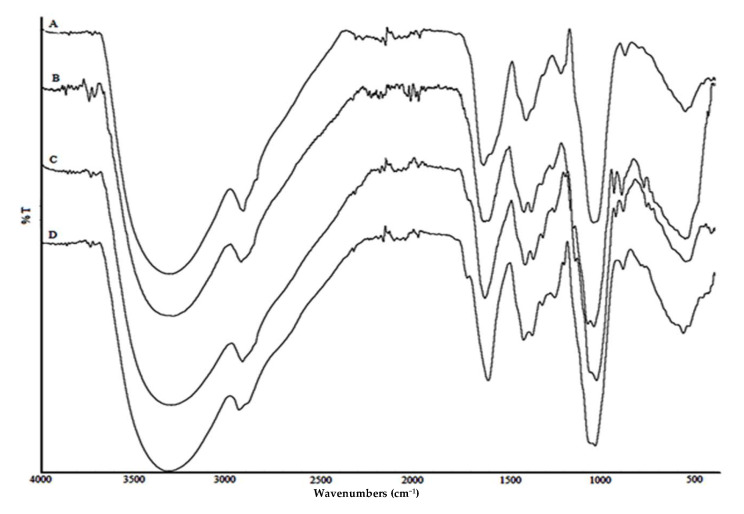
Fourier-transform infrared spectroscopy of polysaccharide samples isolated from *S. commune.* Type of extracts examined: A—filtrate, 14 days, F IT; B—mycelium, 14 days, M IT: C—mycelium, 14 days, M SRB; D—filtrate, 28 days, F SRB.

**Table 1 jof-07-00115-t001:** Comparative review of the antioxidant activity (AO) of submerged polysaccharide (PSH) and ethanol (EtOH) extracts of *S. commune*—28 days—incubation time; DPPH—DPPH assay; ABTS—ABTS assay; TE—Trolox equivalents; FRAP—FRAP assay; AAE—ascorbic acid equivalents.

Extracts		DPPH ^1^ (IC_50_) (µg/mL)		ABTS ^2^(mg TE/g d.w.)		FRAP ^3^(mg AAE/g d.w.)	
*S. commune* IT		PSH	EtOH	PSH	EtOH	PSH	EtOH
F	7 days	**16.03 ± 4.30 ^a^**	72.65 ± 9.15 ^a^	6.64 ± 0.66 ^a^	5.17 ± 0.03 ^a^	65.11 ± 6.27 ^a^	42.58 ± 3.47 ^a^
	14 days	107.94 ± 2.36 ^a^	63.67 ± 2.65 ^a^	**8.20 ± 0.94 ^a^**	4.95 ± 0.41 ^a^	66.73 ± 12.11 ^a^	**99.00 ± 2.73 ^a^**
	21 days	74.76 ± 19.70 ^a^	**55.43 ± 1.89 ^a^**	5.69 ± 0.52 ^a^	**5.27 ± 0.22 ^a^**	**82.96 ± 0.94 ^a^**	91.65 ± 0.96 ^a^
	28 days	189.81 ± 7.96 ^a^	69.15 ± 3.98 ^a^	2.86 ± 1.30 ^a^	3.38 ± 0.06 ^a^	28.44 ± 7.72 ^a^	47.82 ± 5.28 ^a^
M	7 days	94.89 ± 8.41 ^a^	82.69 ± 1.93 ^a^	5.16 ± 1.63 ^a^	5.29 ± 0.05 ^a^	40.03 ± 3.41 ^a^	66.39 ± 3.84 ^a^
	14 days	**14.45 ± 6.83 ^a^**	**49.34 ± 0.65 ^a^**	1.63 ± 1.71 ^a^	5.20 ± 0.08 ^a^	6.49 ± 7.40 ^a^	**81.30 ± 1.98 ^a^**
	21 days	85.39 ± 1.61 ^a^	120.63 ± 3.29 ^a^	**7.49 ± 0.67 ^a^**	**5.84 ± 0.06 ^a^**	**79.12 ± 10.77 ^a^**	14.22 ± 1.32 ^a^
	28 days	93.48 ± 0.01 ^a^	105.39 ± 1.66 ^a^	5.12 ± 0.39 ^a^	4.85 ± 0.19 ^a^	33.67 ± 5.35 ^a^	32.46 ± 1.27 ^a^
*S. commune* SRB							
F	7 days	**15.76 ± 0.63 ^a^**	70.86 ± 2.52 ^b^	2.60 ± 0.14 ^a^	3.80 ± 0.08 ^a^	0.28 ± 0.09 ^b^	61.81 ± 4.78 ^a,c^
	14 days	32.42 ± 3.09 ^a^	**53.61 ± 2.32 ^b^**	7.40 ± 0.19 ^a^	3.26 ± 0.96 ^a^	9.69 ± 3.67 ^b^	**107.86 ± 12.8 ^a,c^**
	21 days	30.18 ± 2.25 ^a^	55.96 ± 1.31 ^b^	**7.62 ± 0.62 ^a^**	**4.65 ± 0.31 ^a^**	**57.10 ± 2.35 ^b^**	89.97 ± 12.81 ^a,c^
	28 days	69.69 ± 3.93 ^a^	67.31 ± 1.03 ^b^	3.25 ± 1.98 ^a^	3.40 ± 0.09 ^a^	4.36 ± 1.36 ^b^	80.28 ± 3.21 ^a,c^
M	7 days	**18.92 ± 6.12 ^a^**	377.71 ± 14.85 ^b^	**7.78 ± 0.41 ^a^**	**6.01 ± 0.07 ^a^**	**20.89 ± 1.67 ^b^**	**11.44 ± 0.84 ^a,d^**
	14 days	71.55 ± 3.43 ^a^	**74.65 ± 1.74 ^b^**	3.17 ± 0.44 ^a^	3.02 ± 0.10 ^a^	9.55 ± 0.51 ^b^	5.99 ± 0.20 ^a,d^
	21 days	69.13 ± 4.24 ^a^	164.28 ± 2.52 ^b^	2.30 ± 0.31 ^a^	5.43 ± 0.53 ^a^	5.61 ± 1.83 ^b^	10.77 ± 1.05 ^a,d^
	28 days	51.41 ± 11.27 ^a^	137.5 ± 5.87 ^b^	2.14 ± 0.35 ^a^	5.49 ± 0.23 ^a^	0.90 ± 1.91 ^b^	10.91 ± 0.46 ^a,d^

Each value is expressed as mean ± SD. ^1,2,3^ Means with different letters (**a**–**d**) within two columns per test (DPPH, ABTS and FRAP) are significantly different (Tukey’s HSD, ANOVA, LSD, Friedman). Significant differences between extracts were determined by Tukey’s HSD test and LSD post hoc test at *p* < 0.05. The differences are related to isolates (IT/SRB) and type of extracts (PSH/EtOH), and F/M (second letters). Bold values stand for the most promising antioxidant activities.

**Table 2 jof-07-00115-t002:** Antiradical activity and total phenol content (TP) of submerged EtOH extracts of *S. commune* strains from Italy (IT) and Serbia (SRB). F—filtrate; M—mycelium; 7–28 days—Incubation time; GAE—gallic acid equivalents.

Extracts		SOA (IC_50_) (µg/mL)	OH (IC_25_) (µg/mL)	NO (IC_25_) (µg/mL)	TP (mgGAE/g.d.w.)
*S. commune* IT					
F	7 days	526.68 ± 152.45 ^a^	103.02 ± 12.23 ^a^	614.02 ± 14.79 ^a^	69.48 ± 1.00 ^a^
	14 days	652.14 ± 6.56 ^a^	124.27 ± 7.00 ^a^	965.43 ± 129.75 ^a^	75.23 ± 1.04 ^a^
	21 days	90% ^1^	110.85 ± 2.31 ^a^	**707.05 ± 34.84 ^a^**	**77.52 ± 0.97 ^a^**
	28 days	90% ^1^	**10.80 ± 0.54 ^a^**	1161.41 ± 48.90 ^a^	68.70 ± 0.30 ^a^
M	7 days	602.41 ± 0.01 ^a^	**20.20 ± 1.36 ^a^**	**788.80 ± 36.36 ^a^**	53.49 ± 1.97 ^a^
	14 days	**216.98 ± 45.84 ^a^**	228.36 ± 3.25 ^a^	1182.78 ± 36.35 ^a^	**76.65 ± 1.30 ^a^**
	21 days	634.14 ± 38.25 ^a^	44.29 ± 4.20 ^a^	n.a. ^2^	42.74 ± 1.40 ^a^
	28 days	492.92 ± 125.11 ^a^	84.74 ± 2.36 ^a^	n.a. ^2^	70.01 ± 0.65 ^a^
*S. commune* SRB					
F	7 days	635.74 ± 8.99 ^a^	494.34 ± 59.08 ^a^	1794.48 ± 13.21 ^a^	**84.60 ± 1.64 ^a^**
	14 days	90% ^1^	257.92 ± 0.01 ^a^	<IC_25_ ^3^	81.93 ± 0.81 ^a^
	21 days	90% ^1^	261.38 ± 0.01 ^a^	<IC_25_ ^3^	78.27 ± 1.75 ^a^
	28 days	90% ^1^	**89.69 ± 0.38 ^a^**	<IC_25_ ^3^	76.14 ± 1.42 ^a^
M	7 days	**161.60 ± 2.06 ^a^**	170.57 ± 0.35 ^a^	737.12 ± 32.91 ^a^	40.22 ± 0.82 ^a^
	14 days	253.14 ± 17.33 ^a^	163.94 ± 19.22 ^a^	145.88 ± 2.83 ^a^	**82.62 ± 0.99 ^a^**
	21 days	429.60 ± 44.01 ^a^	89.45 ± 0.01 ^a^	785.34 ± 39.03 ^a^	63.28 ± 1.28 ^a^
	28 days	218.01 ± 13.59 ^a^	**85.57 ± 0.12 ^a^**	**112.19 ± 19.73 ^a^**	2.09 ± 0.06 ^a^

Data are reported as mean ± SD of triplicates. Bold values highlight the best results. ^1^ 90%, inhibitory concentration is higher than IC_50_; ^2^ n.a.—Non-active; ^3^ <IC_25_, the activity of analyzed samples is lower than IC_25_. ^a^ In each column different letters, mean significant differences (Tukey’s HSD test at *p* < 0.05). The difference is related to isolate (IT/SRB) and type of samples (F/M).

**Table 3 jof-07-00115-t003:** Determined concentrations of selected phenolic compounds using LC–MS/MS technique in examined polysaccharide and ethanolic extracts (µg/g.d.w.).

Extracts	Class of Analyzed Compounds			
	Hydroxybenzoic Acids			Cyclohexane Carboxylic Acid
	*p*-Hydroxybenzoic Acid	Protocatechuic Acid	Gallic Acid	Quinic Acid
**PSH**				
IT F 7	**5.50**	**3.17**	<12.2 *	<3.05 *
IT M 7	**1.87**	<1.525 *	<12.2 *	<3.05 *
IT M 14	**8.28**	<1.525 *	<12.2 *	<3.05 *
SRB F 7	<1.525 *	<1.525 *	<12.2 *	**4.29**
**EtOH**				
IT F 14	**3.81**	<1.525 *	<12.2 *	**13.24**
IT F 21	**9.51**	<1.525 *	<12.2 *	<3.05 *
IT M 14	**4.10**	<1.525 *	<12.2 *	**8.57**
IT M 21	**9.66**	<1.525 *	<12.2 *	<3.05 *
IT M 28	**22.19**	**5.41**	<12.2 *	**4.82**
SRB F 7	**12.14**	<1.525 *	<12.2 *	**15.51**
SRB F 14	**3.77**	<1.525 *	75.77	<3.05 *
SRB F 21	**11.25**	**1.90**	<12.2 *	<3.05 *
SRB F 28	**7.69**	<1.525 *	<12.2 *	<3.05 *
SRB M 7	**5.93**	<1.525 *	<12.2 *	**19.52**
SRB M 14	**5.94**	<1.525 *	<12.2 *	**20.06**
SRB M 21	**6.52**	<1.525 *	<12.2 *	**5.49**

**Bold number**: the amount of quantified phenolic compounds in examined extracts. * Number: detected compound—Peak observed, concentration is lower than the LoQ (limit of quantification) but higher than the LoD (limit of detection). PSH—polysaccharide extracts; EtOH—ethanolic extracts; IT—isolate from Italy; SRB—isolate from Serbia; F—filtrate; M—mycelium; 7, 14, 21, 28—days of incubation.

## Data Availability

Not applicable.

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
