# Peer review of "Comparison of Two Schizophyllum commune Strains in Production of Acetylcholinesterase Inhibitors and Antioxidants from Submerged Cultivation"

_jof, 2021, doi:10.3390/jof7020115_

Round 1

Reviewer 1 Report

This is an interesting article suggesting that schizophyllan has anti ACHE activity. The authors should show that this is actually the case. Schizophyllan can be identified by NMR or by analyzing beta-glucanase products. The manuscript is not easy to read.

Other points

  1. Grammar and spelling should be checked throughout the manuscript by a native speaker
  2. Reference 18 should be replaced by another reference such as Ohm et al., 2010 Nature Biotechnology to refer to the occurrence of S. commune in nature as a model organism of mushroom production.
  3. How was identification of S. commune be performed ? this is not described.
  4. Are strains of the Fungicult collection publically available ?
  5. Are the strains mono- or dikaryons ?
  6. Line 93-108. This part of the Methods is very poorly described and cannot be repeated by other scientists
    1. Were cultures grown in the light or dark; what kind of bottles or Erlenmeyers were used for liquid cultures ?
    2. Was mycelium dried before making a fine powder ?
    3. What were the modifications that were used based on Ren et al. ?
  7. Line 128 and others; introduce the abbreviations the first time it is used for instance FRAP ?
  8. The PSH fraction is important for this article; yet the author does not know what it is. This is strange ! Is it schizophyllan ?
  9. Figure S1 and S2 lack SD’s and it is not explained from how much medium the mass was derived.
  10. Line 314; two among tested ?
  11. Line 464; the authors have not shown that the geographical origin affects the activity of the strains; this can be coninciden

Author Response

Dear Reviewer, 

Reviewer 2 Report

MS titled “Comparison of two Schizophyllum commune strains in production of AChE inhibitors and antioxidants from submerged cultivation” is presented interesting results in field of mushroom biotechnology. I recommended article for publishing in Journal Of Fungi after minor correction.
My suggestions:
1. Title: the abbreviation AChE should be change to whole name.
2. Please to check the MS, because in many place in text are double space (for example line:56; 62; 97, 104; 112; 211 etc.)
3. 62 Line: “..strong antitumor activity..” what means strong in my opinion this word is not scientific.
4.117 Line: please to add λ= 412; in the sentence: The absorbance was…
5. References: citation number 33 please to improve this one.

Author Response

Dear Reviewer, 

please see the attachement.
